# Circulating Small EVs miRNAs as Predictors of Pathological Response to Neo-Adjuvant Therapy in Breast Cancer Patients

**DOI:** 10.3390/ijms232012625

**Published:** 2022-10-20

**Authors:** Oana Baldasici, Loredana Balacescu, Daniel Cruceriu, Andrei Roman, Carmen Lisencu, Bogdan Fetica, Simona Visan, Andrei Cismaru, Ancuta Jurj, Lucian Barbu-Tudoran, Valentina Pileczki, Laurian Vlase, Oana Tudoran, Ovidiu Balacescu

**Affiliations:** 1Department of Genetics, Genomics and Experimental Pathology, The Oncology Institute “Prof. Dr. Ion Chiricuță”, 400015 Cluj-Napoca, Romania; 2Department of Pharmaceutical Technology and Biopharmaceutics, “Iuliu Haţieganu” University of Medicine and Pharmacy, 400347 Cluj-Napoca, Romania; 3Department of Molecular Biology and Biotechnology, “Babes-Bolyai” University, 5-7 Clinicilor Street, 400006 Cluj-Napoca, Romania; 4Department of Radiology, The Oncology Institute “Prof. Dr. Ion Chiricuță”, 400015 Cluj-Napoca, Romania; 5Department of Pathology, The Oncology Institute “Prof. Dr. Ion Chiricuță”, 400015 Cluj-Napoca, Romania; 6Research Center for Functional Genomics, Biomedicine and Translational Medicine, “Iuliu Haţieganu” University of Medicine and Pharmacy, 400347 Cluj-Napoca, Romania; 7Electron Microscopy Center “Prof. C. Craciun”, Faculty of Biology & Geology, “Babes-Bolyai” University, 5-7 Clinicilor Street, 400006 Cluj-Napoca, Romania; 8Electron Microscopy Integrated Laboratory, National Institute for R&D of Isotopic and Molecular Technologies, 67-103 Donat Street, 400293 Cluj-Napoca, Romania

**Keywords:** breast cancer, neo-adjuvant therapy, pathological response, liquid biopsy, plasma small EVs, exosome, miRNA, biomarker, prediction

## Abstract

Neo-adjuvant therapy (NAT) is increasingly used in the clinic for the treatment of breast cancer (BC). Pathological response to NAT has been associated with improved patients’ survival; however, the current techniques employed for assessing the tumor response have significant limitations. Small EVs (sEVs)-encapsulated miRNAs have emerged as promising new biomarkers for diagnosis and prediction. Therefore, our study aims to explore the predictive value of these miRNAs for the pathological response to NAT in BC. By employing bioinformatic tools, we selected a set of miRNAs and evaluated their expression in plasma sEVs and BC biopsies. Twelve miRNAs were identified in sEVs, of which, miR-21-5p, 221-3p, 146a-5p and 26a-5p were significantly associated with the Miller–Payne (MP) pathological response to NAT. Moreover, miR-21-5p, 146a-5p, 26a-5p and miR-24-3p were independent as predictors of MP response to NAT. However, the expression of these miRNAs showed no correlation between sEVs and tissue samples, indicating that the mechanisms of miRNA sorting into sEVs still needs to be elucidated. Functional analysis of miRNA target genes and drug interactions revealed that candidate miRNAs and their targets, can be regulated by different NAT regimens. This evidence supports their role in governing the patients’ therapy response and highlights their potential use as prediction biomarkers.

## 1. Introduction

Early detection through mammographic screening and improved treatment options have increased BC survival in westernized countries [1]. However, around 30% of cancer patients fail to respond to conventional treatments, leading to tumor recurrence. The main prognostic factors associated with BC are the number of involved lymph nodes, tumor size, histological grade, and hormone receptor status, while the treatment decision is based on both clinical and molecular characteristics of the tumor [2]. NAT is generally the first option before surgical treatment of locally advanced tumors, as it can downstage the extent of the disease, increase the rate of breast-conserving therapy [3,4], and offer information regarding the tumor response to systemic therapy. This information can guide treatment decision in adjuvant settings and provide prognostic data regarding patients survival outcomes [5]. Tumor response to systemic therapy is evaluated by clinical and radiological examination followed by a histological assessment of the excised tumor tissue [1]. However, clinical evaluation is often under or overestimated, thus pathological examination of the tumor bed following NAT is the gold standard, being associated with overall and disease-free survival [6]. Consequently, several pathological evaluation systems have been proposed in order to better stratify patients’ prognoses, including MP system [7] and residual cancer burden (RCB) index [8]. The MP method comprises a five-grade pathological evaluation system, with grade 5 representing pathological complete response (pCR), with no malignant cells identifiable in sections from the site of the tumor and grades 1–4 representing partial responses, with variable reductions 0–90% in tumor cellularity [7]. Besides cellularity, the RCB index takes into account additional variables such as the bidimensional diameter of the tumor, lymph node status, and metastasis. The pCR using RCB index is defined as RCB-0 [8]. Both methods have shown associations with overall and disease-free survival, highlighting them as reliable methods for evaluating the pathological tumor response to neoadjuvant therapy [9]. However, the histological assessment of NAT response can be achieved only at the time of surgery, after the patient has been exposed to several rounds of treatment. Therefore, the identification of predictive biomarkers of pathological response that could be evaluated before therapy onset is vital for stratifying patients that are more likely to benefit from NAT and may prevent over or undertreatment.

microRNAs (miRNAs) have been described to play a significant role in cancer [10]. Numerous reports of altered miRNA expression have been linked to various pathologies [11,12,13], including BC [14,15]. MiRNAs are small non-coding RNAs that regulate gene expression in key biological processes such as proliferation, apoptosis, invasion, and metastasis [16,17,18] and have recently emerged as regulators of the complex mechanisms that drive BC therapeutic resistance or sensitivity. Extracellular vesicles (EVs) are small lipid bilayer-encapsulated vesicles that are released by living cells in the microenvironment. Based on their size, EVs can be classified into two main subtypes: ectosomes (100–1000 nm) and small extracellular vesicles (sEVs, 30–150 nm, also referred to as exosomes by the literature), that encapsulate various components from the cell of origin, such as small RNA species and proteins [19]. Recent data emphasized that sEVs which carry miRNAs possess higher stability in various body fluids and can be selectively delivered to tissues based on surface markers [20]. Thus, tumor cells shed miRNAs into the circulation which are delivered to and up-taken by recipient cells in adjacent or distant tissues [21]. This horizontal transfer of miRNA functions as endogenous signaling mechanisms to reshape the microenvironment as tumor favoring–drug resistant niches [22]. Previous studies suggest that circulating sEVs miRNAs represent specific and stable non-invasive molecular biomarkers in disease progression and treatment efficacy [23,24,25]. Few studies have investigated the predictive power of miRNAs in pathological response following NAT in BC. Most of them were only focused on triple negative BC (TNBC) [26,27,28], and were evaluated in tissue samples [29], or enrolled a smaller patient cohort [9,30]. Considering that sEVs miRNAs detected in liquid biopsies represent more accessible and less invasive biomarker sources, our study aims to explore the predictive value of these miRNAs for the pathological response to NAT in BC.

## 2. Results

### 2.1. Patients and Tumor Characteristics

The clinical-pathological data of the patients included in the study are summarized in Table 1. The patient cohort consisted of 43 patients with matched biopsies and plasma samples, for 29 patients only plasma samples were available, while for 10 only biopsy samples. The median age of the patients included in the study was roughly 61, most of the patients being over 50 years old at the time of diagnosis. Around 50% of the patients had moderately differentiated carcinomas, with higher than 20 proliferation index. Most of the patients had luminal tumors, over 85% received an advanced-stage diagnosis (>II). Of the patients that received neoadjuvant therapy (NAT), around 75% received only chemotherapy, 16–17% received only hormonal therapy, while the rest received combinatory regimens of chemotherapy and hormonal therapy. One patient with disease progression also received radiotherapy in combination with chemotherapy and hormonal therapy. TNM classification system of malignant tumors was retained for prognostic information (primary and post-NAT surgery), while MP and RCB systems were used to evaluate the pathological response of the patients to NAT. According to the MP evaluation, around 30% of the patients had no response, 11% presented minor response, less than 30% presented intermediate response, while over 25%, had almost complete pathological response. RCB system classification data revealed that only 15% of the patients reached pathological complete response, almost a quarter were therapy resistant, while half had partial responses.

### 2.2. Selection of miRNAs of Interest

GEO2R analysis of the GSE25066 dataset revealed a list of 250 genes (−1.5 > FR > 1.5, BH adjusted *p*-value < 0.05) differentially expressed between the resistant and responsive BC patients. MiRwalk analysis predicted this list to be targeted by 957 individual miRNAs. NCI-60 revealed 60 miRNAs associated with paclitaxel and 20 associated with doxorubicin responses (*p*-value < 0.05, Pearson > 0.85), while 60 miRNAs were previously validated as paclitaxel and doxorubicin targets according to Pharmaco-miR database (Appendix A). Of these, we randomly selected and further investigated in plasma sEVs, 33 miRNAs as predictive biomarkers for therapy response. (Table 2).

### 2.3. Investigation of microRNA Expression in Plasma sEVs

Of the 33 investigated miRNAs, only 12 miRNAs presented clear exponential amplification curves, 10 miRNAs had low or unspecific amplification and 11 miRNAs were negative for amplification. The qPCR amplification efficiency for miRNA advanced assays was 1.935 (96.75%). The statistical data association between the 12 quantifiable sEVs miRNAs expression and the clinic-pathological data of the patients included in the study are presented in Table 3. No correlations were observed between the investigated miRNA expression and patients age, while only slight associations were observed between miRNAs expression and the rest of the clinical-pathological data. Of note, significant different expression was calculated for miR-143-3p, 146a-5p, 24-3p, 193b-3p, 92a-3p, 484, 185-5p and 26a-5p and clinical lymph node status. Most notably, miR-21-5p, 221-3p, 146a-5p and 26a-5p were significantly associated with the MP pathological response to NAT; in general, their expression is decreased in patients that have improved response compared to the ones that are resistant (Figure 1). No significant associations were observed for the RCB evaluation system.

### 2.4. Investigation of microRNA Expression in Tissue Biopsies

Four miRNAs were associated with MP pathological response and further investigated for their expression in BC tissues (Table 4). Of note, only miR221-3p and miR-146a-5p were significantly associated with pathological response to NAT, higher expression was observed in tissues with pathological complete response (G5) compared to partial (G3) or non-responsive samples (G1) (Figure 2). None of the miRNAs expression correlated between the biopsy and sEVs samples. (Table 4). Analysis of the miRNAs expression in the TCGA database was performed as an independent external cohort (Appendix A).

### 2.5. Prediction Analysis of miRNA Expression in Relation with Pathological Response

Ordinal logistic regression was used to test the prediction value of the four miRNAs of interest in pathological response to NAT, in both tissue and plasma samples. We found the expression of circulating sEVS miR-21-5p (OR = 0.603, 95% CI = 0.386–0.986, *p* = 0.044), miR-146a-5p (OR = 0.144, 95% CI = 0.026–0.799, *p* = 0.027), miR-24-3p (OR = 0.019, 95% CI = 0–0.953, *p* = 0.047) and miR-26a-5p (OR = 0.407, 95% CI = 0.189–0.874, *p* = 0.021) expression to be independent predictors of MP response. In general, for every unit increase in miRNA expression, there is a decreased probability in the odds of patients having an improved pathological response. In multivariable analysis with the full set of the four miRNAs as independent variables, the fitted model showed significant improvement over the null model (*p* = 0.011); however, none of the independent variables were found to have a significant effect on the model. No significant predictors were calculated for the RCB response. Tissue evaluation showed no significant predictive value for the four miRNAs set.

### 2.6. Functional Analysis of miRNA Target Genes and Drug Interactions

In order to provide a biological context to our results (for our four miRNAs of interest), miRTarBase was interrogated for validated mRNA targets. In parallel, mRNA and miR-21, miR-221, miR-146a and miR-26a expression data from TCGA were correlated in the matching samples. The two lists were intersected to generate lists of validated miRNAs—mRNA targets, considering of interest only the anticorrelated genes (r < −0.2, *p* < 0.05). Fifty-seven genes were identified to be anticorrelated with miR-21, sicty-nine genes with miR-221, eighteen genes with miR-146a and six genes with miR-26a (Appendix A). These lists were imported in miRWalk3 for subsequent characterization using the GSEA built-in option (Table 5). Gene ontology analysis depicted the association of the miRNAs of interest with the biological processes, molecular function and cellular components associated with the significant target genes. Thus, through their anticorrelated target genes, miR-21-5p was mainly associated with kinase and ubiquitin transfer activities. miR-221-3p targets were associated with protein processing such as ubiquitination, destabilization and peroxisome targeting and regulation of the proliferation and apoptotic processes possibly through transcription regulation. A reduced number of anticorrelated genes were identified for the miR-146a and miR-26a, the only significant processes identified were that miR-146a targets are part of the receptor complex compartments, while miR-26a targets belong to the glutamatergic synapse compartment indicating a kinase activity. While no significant pathways were identified using the KEGG (Kyoto Encyclopedia of Genes and Genomes) analysis option, Reactome database revealed that miR-221-3p target genes are associated with protein localization and gene transcription pathways.

In order to comprehensively depict the role of the 4 miRNAs in drug response regulation, we interrogated the pharmaco-miR database for known drug-miRNA-mRNA interactions (Table 6). Through their mRNA targets, miR-21-5p, miR-221-3p and miR-146a-5p have been found to be known mediators of BC chemotherapeutics such as 5-fluorouracil, taxanes, cisplatin or doxorubicin, or targeted therapeutics such as tamoxifen and trastuzumab.

## 3. Discussion

Treatment response represents one of the most significant issues of BC management, since about 30% of BC patients develop resistance to the standard therapeutic scheme, leading to cancer progression and decreased survival [31]. Locally advanced BC patients are subjected to neo-adjuvant therapy in order to reduce the tumor size and improve the surgical decision (the ability to choose breast-conserving surgery) as well as a method for assessing the patient’s response to systemic therapy [32]. Recent data suggest that pathological complete response (pCR) after NAT is associated with better patient prognosis [6]. However, the pathological response to NAT can only be assessed at surgery, after several cycles of systemic therapy, exposing the non-responding patients to unnecessary toxicity. Significant efforts have been made for developing gene-expression based predictors which could better stratify patients that could benefit from chemotherapy or hormonal treatment (Oncotype Dx, MammaPrint) and anticipate their prognosis (MapQuant D, Endopredict, the BC Index (BCI), and PAM50-ROR) [33,34]. Even if these tests have proven their usefulness, they are difficult to implement in the standard clinical routine due to their high cost and infrastructure requirements. Moreover, these tests demand fresh tissue samples, which can be difficult to harvest and involve invasive procedures for the patient. Therefore, the identification of novel, less invasive and more easily-accessible biomarkers for predicting BC response to the available therapeutic scheme is essential for globally improving BC management.

Liquid biopsy emerged as a complementary method for investigating molecular biomarkers in various body fluids, that could be used to assess the dynamic phenotype of BC patients. Together with the increasing interest for liquid biopsy, tumor derived sEVs miRNAs arose as a promising new class of biomarkers for cancer progression and treatment efficacy [23,24,25,35]. Circulating exosomes are crucial mediators of intercellular communication, their cargos mirroring the originating cells physiological state. By transferring cargos between tumor cells and adjacent or distant sites, exosomes reshape the microenvironment to establish tumor favoring–drug resistant niches [36]. Therefore, this study aimed to determine the predictive value of circulating sEVs miRNAs regarding pathological response to NAT and their expression correlation with the clinico-pathological data of BC patients. While several evaluation methods have been established [37], herein, we used RCB index and the MP system to evaluate pathological response to NAT. Both systems have been shown to accurately stratify survival outcome of patients after NAT [9].

By employing multiple publicly available databases, we explored the associations between miRNA expression and therapy response, drug interactions and their predicted mRNA targets, and identified 33 miRNAs of interest of which 12 could be accurately quantified in BC circulating sEVs (Table 3).

Two sEVs miRNAs, miR-328-3p and miR-34a-5p that have been previously reported as predictive biomarkers of NAT response in BC [30], had low or no expression in our samples. Moreover, plasma circulating miRNAs that have been described to be associated with pathological response in Her2+ BC patients (rev in [38]) such as miR-140-3p, miR-195-5p, miR- 373-3p, and let-7a-3p were negative in our samples. Furthermore, plasma miR-375-3p association with NAT response is well documented [39,40], suggesting that the presence of these miRNAs in plasma samples is not as an EVs cargo. However, the inconsistencies between studies, could be related to the use of different isolation, quantification and normalization methods of circulating miRNA in plasma in general and exosomes and EVs in particular. As no optimal normalization strategy for exosomal and sEVs miRNA expression is consensually accepted so far, variations in data interpretation and biological predicted effects are possible, leading to limitations in comparing research studies, and resulting in misleading conclusions [41]. Thus, validation of miRNA measurement across laboratories still remains a stringent future aspiration.

Four circulating sEVs miRNAs, notably miR-21-5p, 221-3p, 146a-5p and 26a-5p were found to be significantly associated with the MP pathological response to NAT. When their expression was evaluated in tissue samples, only miR-221-5p and miR-146a-5p retained statistical significance. However, no correlation was observed, when we compared the miRNA expression levels between plasma sEVs and tissue samples.

Previous data pointed out that the miRNAs loading into exosomes is not a passive process, and the repertoire of miRNAs content into exosomes may differ from that of the producer cell [42], assorting of oncomiRs into exosomes is generally suppressed, while the sorting of tumor suppressor miRs (TS-miRs) into exosomes is increased [43].

Three of these miRNAs, mir-21-5p, miR-146a-5p and miR-26a -5p were found to be individual independent predictors of MP pathological response in ordinal logistic regression, higher sEVs expression predicting decreased odd ratios of reaching pathological complete response. Consistently with these results, sEVs expression of these miRNAs was significantly lower in patients with MP partial response (G3) compared to non-responding ones (G1). Moreover, lower sEVs miR-21-5p expression was associated with MP pathological complete response (G5).

MiR-21-5p is a well-established oncomiR, that has been associated with chemoresistance in various cancers, such as lymphoma, colorectal, ovarian, lung and BC (rev in [44]). The literature data regarding the role of miR-21-5p in BC are conflicting. Studies reported both increased and decreased expression associations with NAT response. Higher miR-21-5p expression in plasma exosomes was directly correlated with tumor size in patients undergoing chemotherapy [45] and increased expression in patient’s serum after NAT was predictive of overall survival (OS) [46]. On the other hand, Liu et al. [47] reported decreased miR-21-5p serum expression associated with clinical response to NAT, patients with increased levels of serum miR-21-5p during NAT had a worse disease-free survival than those with decreased levels (rev in [48]), while miR-21-5p overexpression on tumor tissues induced resistance to neoadjuvant therapy combined with Trastuzumab and chemotherapy [49]. We have found that lower circulating sEVs miR-21-5p levels are associated with better pathological response to NAT and it is an independent predictor of MP response, consistent with previous data reported by McGuire et al. [50]. Additionally, based on miRNA-mRNA correlation and target prediction data, we estimate that miR-21-5p has a role in protein kinase and transferase activity. MiR-21-5p has been previously reported to regulate chemosensitivity to classic BC treatment regimens by targeting tumor-suppressors such as *PTEN* and programmed Cell Death 4 (*PDCD4*) [51,52,53] in various cancers.

Conflicting data regarding the oncogenic or tumor suppressive role of miR-221-3p in different cancers indicate its dual role in tumorigenesis [54,55]. In BC, high expression of miR-221-3p has been related to lymph node metastasis, distant metastasis, and poor prognosis [56]. Consistently, our results show increased circulating sEVs miR-221-3p in positive lymph node patients and worse pathological response to NAT. However, these data are conflicting with those from tumor tissue, where higher miR-221-3p was associated with complete response (G5) compared to partial (G3) and non-responders (G1). Even so, our results are confirmed by the study of Hanna et al. [57], that revealed that patients with higher level of tissue miR-221-3p had better survival prognosis. GSEA analysis suggests that through its molecular targets, miR-221-3p acts as a transcription regulator in processes such as proliferation and apoptosis. MiR-221 has been reported to downregulate some tumor suppressor pathways such as *PTEN/Akt/mTOR* signaling and thus promote chemoresistance of BC cells [58]. Moreover, miR-221/222 cluster has been shown to regulate *ER-α* [59], and its suppression has been shown to sensitize BC cells to tamoxifen [60]. A phase I clinical trial with antisense oligonucleotide targeting miR-221-3p (NCT04811898) is currently undergoing to assess the safety and tolerability in advanced solid tumor patients.

The predictive role of miR146a-5p has been studied in various cancers. Higher levels of miR-146a-5p in serum and plasma have been associated with worse response to platinum-based treatment in lung cancer [61,62], Imatinib therapy in patients with chronic myeloid leukemia (CML) [63], hormone [64] and radio [65] therapy in prostate cancer. In contrast, in acute myeloid leukemia (AML), high miR-146a-5p expression and/or upregulation of this miRNA during granulocyte-colony stimulating factor (*G-CSF*) priming chemotherapy was predictive of better clinical outcomes [66]. Similarly, advanced non-small cell lung cancer (NSCLC) patients with low serum exosomal miR-146a-5p levels had higher recurrence rates than those with high levels, suggesting that miR-146a-5p increased the chemosensitivity of NSCLC to cisplatin [67]. The role of miR-146a-5p in BC remains largely unexplored. We found sEVs miR-146a-5p to be an independent predictive factor in MP pathological response higher sEVs miR-146a-5p expression in plasma patients being associated with worse response to NAT. Of interest, in tissue samples, higher miR-146a-5p expression were observed in nonresponsive patients (G1, G2) when compared to partial responders (G3), but also in patients with complete response (G5) compared to those with partial response (G3). Mechanistically, exosomal miR-146a-5p has been shown to confer prostate cancer cells metastatic abilities through *EGFR/ERK* pathway [65], downregulate BRCA1 in triple negative sporadic BCs [68] and play a central role within the *STAT/IFN* signaling axis to create an immunosuppressive microenvironment [69,70].

Consistent with our results, miR-26a-5p, has been previously shown to be involved in pathways related to drug sensitivity/resistance [71], being associated with poor prognosis and predictive of the therapeutic response [72,73]. MiR-26a-5p expression is regulated by a negative estrogen signaling feedback loop [72,73] and has been shown to be a trastuzumab-inducible microRNAs [74] that seems to play an important role in resistance to trastuzumab therapy [75].

## 4. Materials and Methods

### 4.1. BC Patients and Samples Collection

Eighty-two female patients diagnosed with invasive BC at The Oncology Institute “Ion Chiricuță”, Cluj-Napoca, Romania (IOCN) gave their written consent for participation in this study in accordance with the Declaration of Helsinki. The study was approved by the IOCN ethical committee (Approval No. 59/29 November 2016). Core biopsies (*n* = 53) and peripheral blood (*n* = 72) were collected at the time of diagnostic procedure.

### 4.2. Plasma Processing and RNA Extraction from sEVs

Peripheral blood from 72 BC (BC) patients was collected in EDTA collection tubes and processed for plasma separation by double centrifugation at 4000 and 12,000 rpm for 10 min at 4 °C. The samples were aliquoted and stored at −80 °C until sEVs RNA extraction. Small EVs isolation was conducted on 400 μL prefiltered plasma following the Total Exosome Isolation Kit from plasma (Thermo Fisher, Waltham, MA, USA) protocol. Our preliminary data (in Appendix A) indicated that including a supplementary step of filtration (0.8 μm filter) considerably improves the sample purity, by maintaining the small EVs and by eliminating the large EVs, as also demonstrated by Enderle et al. [76], for a column-based method. Briefly, 400 μL of 0.8-μm-filtered plasma was treated with 0.05 volume of Proteinase K and incubated at 37 °C for 10 min followed by a 30 min incubation at 4 °C with 120 ul of precipitation reagent. Small EVs were separated by centrifugation at 10,000× *g*, resuspended in 200 ul PBS and processed for RNA isolation with Total Exosome RNA and Protein Isolation Kit (Thermo Fisher, Waltham, MA, USA) according to the manufacturer’s instructions 2.5 μL of exogenous cel-miR-39 (2 × 10^8^ transcripts) were used as spike-in for each sample for normalization of miRNA expression.

### 4.3. Biopsies Processing and RNA Extraction

Fifty-three frozen biopsies were homogenized in TriReagent Solution (Ambion, Thermo Fisher Scientific, Waltham, MA, USA) using a Miccra D-1 (Miccra GmbH, Mullheim, Germany) polytron and processed for total RNA extraction using the classic phenol-chloroform method. The RNAs were quantified using NanoDrop ND-1000 (Thermo Scientific, Wilmington, DE, USA) and 2100 Bioanalyzer (Agilent Technologies, Santa Clara, CA, USA).

### 4.4. miRNA Selection

Three approaches were employed to explore for putative miRNA biomarkers of NAT pathological response. First, Gene Expression Omnibus (GEO: https://www.ncbi.nlm.nih.gov/geo/ (accessed on 17 April 2019)) was interrogated for gene expression data of BC patients that underwent neoadjuvant therapy and had documented data regarding the pathological response to treatment. Complete gene expression data for primary breast tissue from a cohort of 508 patients were identified under accession number GSE25066. GEO2R analysis software was used to compare for gene expression data between resistant- RCB III (*n* = 110) and responsive- RCB 0/I (*n* = 118) patients. Fold regulation (FR) of ±1.5 and Benjamini–Hochberg (BH) adjusted *p*-value < 0.05 was considered as significance threshold. Retrieved differentially expressed gene list was imported into the miRwalk webtool and the miRTarBase filter was used to identify validated miRNAs known to target the 3′-UTR, CDS or the 5′-UTR of identified genes. Second, Cellminer Cross DataBase (https://discover.nci.nih.gov/rsconnect/cellminercdb/ (accessed on 17 April 2019)) was used to retrieve BC cell line pharmacogenomic miRNA data in response to classic BC treatments. The Compare Patterns tool was used to compute Pearson correlations between the NCI-60 miRNA BC datasets and paclitaxel and doxorubicin treatments. Last, Pharmaco-mir database was explored for known miRNA-drug interactions. Of interest were miRNAs targets of paclitaxel and doxorubicin and their clinical derivatives.

### 4.5. miRNA Expression Evaluation

Considering that plasma samples contain low quantities of sEVs-miRNA, we used the one-step advanced miRNA system to simultaneously assess the expression levels of multiple miRNAs from the same sample. An amount of 4 μL of plasma sEVs RNAs or 10 ng of total RNA from biopsies were pre-amplified using universal RT miRNA primers to generate cDNAs following the TaqMan Advanced miRNA cDNA Synthesis Kit protocol. Next, 1:10 *v*/*v* diluted cDNAs and specific miRNA advanced assays were amplified with TaqMan Fast Advanced Master Mix (2X) using the LC480 device (Roche, Basel, Switzerland) with the following PCR settings: 55 °C for 2 min to remove RNA contaminants; 95 °C for 20 sec for Taq polymerase amplification; 40 cycles of 95 °C for 3 sec followed by 60 °C for 30 sec for PCR amplification. The efficiency of qPCR amplification was evaluated using the standard curve generated for cel-miR-39 exogenous normalizer, started with 2 × 10^8^ transcripts. ∆∆Ct method was used for miRNA relative quantification by reporting the Ct values of the miRNAs of interest to miR-16-5p Ct values. For the sEVs samples, all Ct values were beforehand normalized to cel-miR-39 expression.

### 4.6. TCGA Data Analysis and Bioinformatics

The Cancer Genome Atlas Breast Invasive Carcinoma (TCGA-BRCA) expression data (miRNA and mRNA) and their clinical information were obtained from National Cancer Institute Genomic Data Commons (NCI GDC) data portal (https://portal.gdc.cancer.gov/ (accessed on 17 April 2019)) and cBioPortal for Cancer Genomics (https://www.cbioportal.org/ (accessed on 17 April 2019)). The miRNA-seq data expressed as reads per million and fragments per kilobase millions mRNA-seq data were filtered and log2 (x + 1) transformed. After processing, a miRNA tissue dataset containing 916 tumoral samples and a mRNA tissue dataset of 983 tumoral samples were retained for subsequent analysis. Spearman correlation was used to test potential miRNAs—mRNA associations and intersected with validated miRNA-target interactions retrieved from miRTarBase.

### 4.7. Gene Set Enrichment and Drug Prediction Analysis

The GSEA built-in option of the miRWalk3 webtool was used to identify possible pathways, biological processes, cellular compartments and molecular functions of the miRNAs of interest. Each list of mRNAs that were anticorrelated with the miRNAs of interest in the TCGA cohort was analyzed for enrichment in the Reactome and KEGG Pathways or Gene Ontology (GO). MultiMiR package in R was used to interrogate Pharmaco-miR database for interactions between possible drugs and the miRNAs of interest and their mRNA targets.

### 4.8. Statistical Analysis

The association between clinico-pathological characteristics and miRNAs expression was evaluated with the Mann–Whitney U test for two categorical variables or the Kruskal–Wallis test, followed by Dunn’s multiple comparison post hoc test in the case of three or more categorical variables based on the data distribution. Fold regulation (FR) was calculated as the ratio between mean value of the interest group and the reference group. Spearman correlation was used to test for correlations between the miRNA expression in sEVs and tissue. Univariate and multivariable ordinal regression analysis was used to test for pathological response miRNA predictors using SPSS (IBM SPSS Statistics for Macintosh, Version 28.0; IBM Corp Armonk, NY, USA). All analyses were considered significant at *p*-value less than 0.05.

## 5. Conclusions

In summary, we investigated the predictive power of microRNAs from plasma sEVs of BC patients, and identified four miRNAs (miR-21-5p, 221-3p, 146a-5p and 26a-5p) that were associated with MP pathological response after neo-adjuvant therapy, of which, three acted as independent predictors (miR-21-5p, 146a-5p and 26a-5p). Bioinformatic analysis revealed that candidate miRNAs, and subsequently their target genes, can be regulated by different NAT regimens. This evidence supports their role in governing the patients’ therapy response and highlights their potential use as prediction biomarkers. The expression levels of these miRNAs in tissue and plasma sEVs s were not correlated, indicating that further functional studies to understand the specific sorting of sEVs miRNAs are needed to fully elucidate the underlying mechanisms involved in shaping the NAT response.

## Figures and Tables

**Figure 1 ijms-23-12625-f001:**
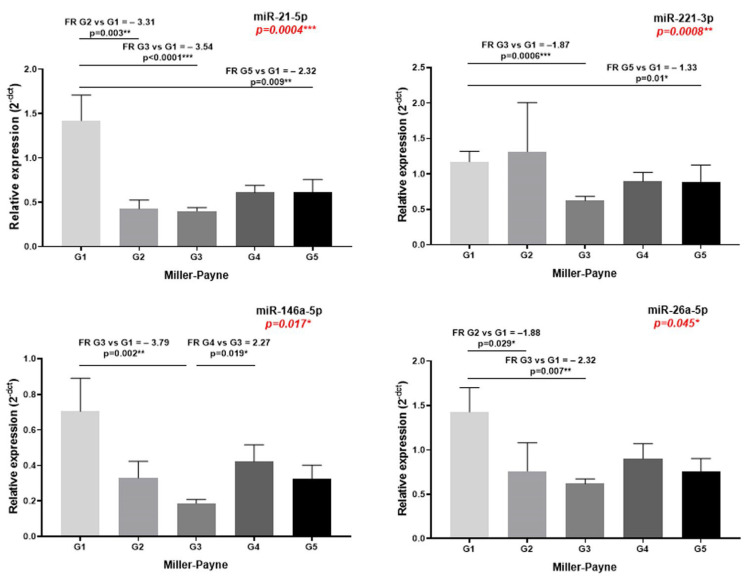
The miR-21-5p, 221-3p, 146a-5p and 26a-5p expression in plasma-derived sEVs, according to MP staging. The differences between groups were evaluated with Kruskal–Wallis test, and pairwise comparisons were performed with Dunn’s test (* *p* < 0.05, ** *p* < 0.01, *** *p* < 0.001).

**Figure 2 ijms-23-12625-f002:**
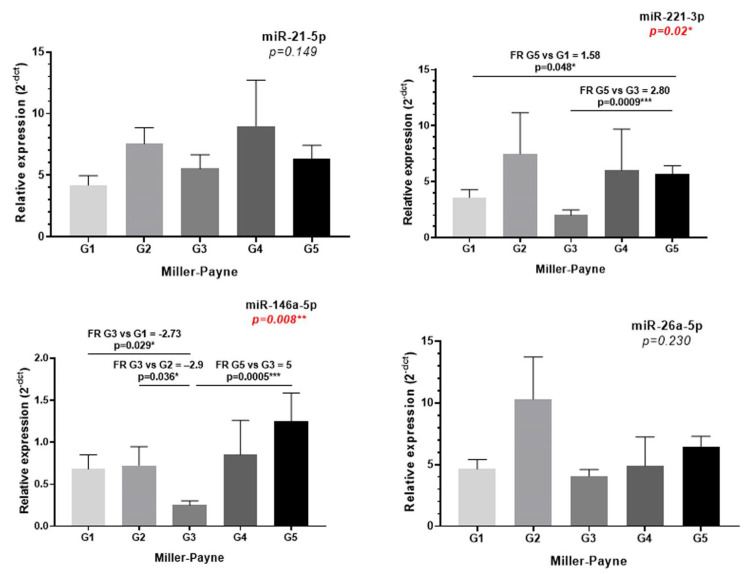
The miR-21-5p, 221-3p, 146a-5p and 26a-5p expression in tissue samples, according to MP staging. The differences between groups were evaluated with Kruskal–Wallis test, and pairwise comparisons were performed with Dunn’s test (* *p* < 0.05, ** *p* < 0.01, *** *p* < 0.001).

**Table 1 ijms-23-12625-t001:** Clinico-pathological data of the patients included in the study.

Variable	Plasma Study	Tissue Study
	*n* = 72	*n* = 53
**Age**		
*Median (range)*	61.5 (35–76)	61 (29–76)
**Grading biopsy**		
*G1*	9 (12.5%)	6 (11.3%)
*G2*	39 (54.2%)	26 (49.1%)
*G3*	24 (33.3%)	21 (39.6%)
**Estrogen Receptor**		
*ER-*	20 (27.8%)	14 (26.4%)
*ER+*	52 (72.2%)	39 (73.6%)
**Progesteron receptor**		
*PR-*	32 (44.4%)	24 (45.3%)
*PR+*	40 (55.6%)	29 (54.7%)
**HER2**		
*HER2-*	64 (88.9%)	47 (88.7%)
*HER2+*	8 (11.1%)	5 (9.4%)
*NA*	-	1 (1.9%)
**KI67**		
≤*20*	31 (43.1%)	20 (37.7%)
*>20*	41 (56.9%)	33(62.3%)
**Molecular Subtype**		
*Luminal A*	23 (31.9%)	15 (28.3%)
*Luminal B*	26 (36.2%)	22 (41.5%)
*LuminalB_HER2+*	3 (4.2%)	1 (1.9%)
*HER2+*	5 (6.9%)	4 (7.5%)
*TNBC*	15 (20.8%)	10 (18.9%)
*NA*	-	1 (1.9%)
**Tumor size (c)**		
*cT1*	4 (5.6%)	2 (3.8%)
*cT2*	36 (50%)	25 (47.2%)
*cT3*	11 (15.3%)	9 (17.0%)
*cT4*	16 (22.2%)	13 (24.5%)
*NA*	5 (6.9%)	4 (7.5%)
**Lymph nodes (c)**		
*cN0*	17 (23.6%)	9 (17.0%)
*cN1*	22 (30.6%)	16 (30.2%)
*cN2*	26 (36.1%)	23 (43.4%)
*cN3*	2 (2.8%)	1 (1.9%)
*NA*	5 (6.9%)	4 (7.5%)
**Metastasis (c)**		
*cM0*	62 (86.1%)	45 (84.9%)
*cM1*	2 (2.8%)	1 (1.9%)
*NA*	8 (11.1%)	7 (13.2%)
**Clinical stage**		
*S-I*	3 (4.2%)	2 (3.8%)
*S-II*	23 (31.9%)	15 (28.3%)
*S-III*	36 (50.0%)	29 (54.7%)
*S-IV*	2 (2.8%)	1 (1.9%)
*NA*	8 (11.1%)	6 (11.3%)
**Tumor size (*p*)**		
*pT0*	11 (15.3%)	7 (13.2%)
*pT1*	22 (30.5%)	21 (39.6%)
*pT2*	28 (38.9%)	19 (35.8%)
*pT3*	2 (2.8%)	2 (3.8%)
*NA*	9 (12.5%)	4 (7.6%)
**Lymph nodes (*p*)**		
*pN0*	29 (40.3%)	25 (47.1%)
*pN1*	20 (27.8%)	14 (26.4%)
*pN2*	12 (16.7%)	8 (15.1%)
*pN3*	4 (5.6%)	3 (5.7%)
*NA*	7 (9.6%)	3 (5.7%)
**Metastasis (*p*)**		
*pM0*	1 (1.4%)	0
*pMx*	58 (80.5%)	44 (83%)
*NA*	13 (18.1%)	9 (17%)
**Lymphatic Invasion**		
*L0*	40 (55.6%)	30 (56.6%)
*L1*	24 (33.3%)	20 (37.7%)
*NA*	8 (11.1%)	3 (5.7%)
	*n* = *62*	*n* = *53*
**Neodjuvant therapy**		
*Only CT*	46 (74.2%)	41 (77.4%)
*Only HT*	10 (16.1%)	9 (17%)
*CT + HT*	5 (8.1%)	3 (5.6%)
*CT + HT + RTE*	1 (1.6%)	-
**MP System**		
*Grade 1*	19 (30.6%)	18 (34%)
*Grade 2*	7 (11.3%)	6 (11.3%)
*Grade 3*	17 (27.4%)	16 (30.2%)
*Grade 4*	6 (9.7%)	4 (7.5%)
*Grade 5*	13 (21.0%)	9 (17%)
**RCB**		
*RCB 0*	10 (16.1%)	8 (15.1%)
*RCB-I*	5 (8.1%)	5 (9.4%)
*RCB-II*	31 (50.0%)	25 (47.2%)
*RCB-III*	16 (25.8%)	15 (28.3%)

ER—estrogen receptor, PR—progesterone receptor, TNBC—triple negative BC, cT—clinic tumor, cN—clinic lymph node, cM—clinic metastasis, pT—pathologic tumor, pN—pathologic lymph node, pM—pathologic metastasis, CT—chemotherapy, HT—hormone therapy, RTE—radiotherapy.

**Table 2 ijms-23-12625-t002:** Candidate biomarker miRNAs selection and evaluation in plasma sEVs.

miRNA	Amplification in Plasma sEVs	GSE25066	Pharmaco-miR	NCI-60
hsa-miR-125b-5p	high	predicted	paclitaxel	doxorubicin
hsa-miR-146a-5p			doxorubicin	doxorubicin
hsa-miR-17-5p				
hsa-miR-185-5p				
hsa-miR-193b-3p				
hsa-miR-21-5p			docetaxel, paclitaxel doxorubicin	
hsa-miR-221-3p				
hsa-miR-24-3p				doxorubicin
hsa-miR-26a-5p				
hsa-miR-484				
hsa-miR-92a-3p				taxol
hsa-miR-143-3p			docetaxel	
hsa-let7c-5p	low	predicted		
hsa-miR-130b-3p			taxol	taxol
hsa-miR-140-3p and 5p				
hsa-miR-18a-5p				
hsa-miR-192-5p				taxol
hsa-miR-328-3p				taxol
hsa-miR-195-5p				
hsa-let-7e-5p				taxol
hsa-miR-31-5p			doxorubicin	
hsa-miR-197-3p	negative	predicted		
hsa-miR-200b-5p				
hsa-miR-203a-3p and 5p			paclitaxel	doxorubicin
hsa-miR-215-5p				
hsa-miR-34a-5p			docetaxel, paclitaxel	
hsa-miR-373-3p				
hsa-miR-520h				
hsa-let-7a-3p			doxorubicin, paclitaxel	taxol
hsa-miR-375-3p				doxorubicin
hsa-miR-589-5p				

**Table 3 ijms-23-12625-t003:** Associations between clinical-pathological features and sEVs miRNA expression.

	miR-21-5p	miR-125-5p	miR-221-3p	miR-143-3p	miR-146a-5p	miR-24-3p	miR-193b-3p	miR-92a-3p	miR-484	miR-185-5p	miR-17-5p	miR-26a-5p
	*p*-Value	*p*-Value	*p*-Value	*p*-Value	*p*-Value	*p*-Value	*p*-Value	*p*-Value	*p*-Value	*p*-Value	*p*-Value	*p*-Value
**Age**	0.592	0.722	0.329	0.982	0.773	0.341	0.904	0.404	0.817	0.110	0.393	0.811
	R = 0.06	R = 0.04	R = 0.12	R = −0.003	R = −0.03	R = −0.11	R = 0.02	R = −0.10	R = −0.03	R = −0.19	R = −0.10	R = −0.03
**Grading Biopsy**	0.193	0.138	0.385	0.215	0.287	0.320	0.529	0.253	0.690	0.418	0.639	0.846
*G3* vs. *G2 vs. G1*												
**Estrogen receptor**	0.820	0.415	0.387	0.802	0.145	0.222	0.722	0.149	0.238	**0.015** *	0.933	0.933
*ER+* vs. *ER−*										FR = −1.21		
**Progesteron receptor**	0.939	0.155	0.378	0.596	0.059	0.064	0.935	0.105	0.228	**0.039** *	0.614	0.828
*PR+* vs. *PR−*										FR = −1.02		
**HER2**	0.156	0.060	0.887	0.316	0.057	**0.044** *	0.782	0.199	0.268	**0.035** *	0.728	0.286
*HER2+* vs. *HER2−*						FR = 1.54				FR = 2.11		
**KI67**	0.686	0.801	0.735	0.608	0.320	0.449	0.204	0.469	0.758	0.378	0.823	0.917
*>20* vs. *≤20*												
**Molecular Subtype**	0.998	0.736	0.641	0.934	0.338	0.519	0.964	0.109	0.435	**0.028** *	0.795	0.938
*TNBC vs.*										FR TNBC vs. LumB = 1.56 **		
*LuminalB* vs. *LuminalA*												
**cT**	0.104	0.799	0.426	0.399	0.208	0.652	0.456	0.986	0.608	0.149	0.530	0.503
*T4* vs. *T3 vs. T1 + T2*												
**cN**	0.060	0.718	0.285	**0.009** **	**0.025** *	**0.027** *	**0.025** *	**0.001** **	**0.005** **	**0.0006** ***	0.054	**0.030** *
*Positive (N1 + N2 + N3)* vs. *Negative (N0)*				FR = −1.47	FR = −1.47	FR = −1.64	FR = −1.69	FR = −2.13	FR = −2.34	FR = −2.19		FR = −2.01
**Clinical Stage**	0.329	0.747	0.715	0.360	0.499	0.233	0.193	0.622	0.540	0.351	**0.037** *	0.794
*High (S3 + S4)* vs. *Low (S1 + S2)*											FR = −1.93	
**pT**	0.341	0.752	0.209	0.755	0.634	0.746	0.127	0.891	0.845	0.448	0.915	0.673
*T2* vs. *T1 vs. T0*												
**pN**	0.286	0.682	**0.024** *	0.614	0.999	0.499	0.537	0.551	0.407	0.615	0.329	0.183
*Positive (N1 + N2 + N3)* vs. *Negative (N0)*			FR = 1.69 *									
**Lymphatic Invasion**	0.637	0.850	0.789	0.843	0.515	0.586	0.909	0.590	0.103	0.166	0.253	0.915
*L1* vs. *L0*												
**MP**	**0.0004** *** FR G5 vs. G1 = −2.32 ** FR G3 vs. G1 = −3.54 **** FR G2 vs. G1 = −3.31 **	0.161	**0.008** ** FR G5 vs. G1 = −1.33 * FR G3 vs. G1 = −1.87 ***	0.174	**0.017** * FR G4 vs. G3 = 2.27 * FR G3 vs. G1 = −3.79 **	0.703	0.175	0.120	0.528	0.178	0.994	**0.045** * FR G3 vs. G1 = −2.32 ** FR G2 vs. G1 = −1.88 *
*G5* vs. *G4 vs. G3* vs. *G2 vs. G1*												
**RCB**	0.303	0.596	0.794	0.686	0.761	0.442	0.423	0.497	0.672	0.097	0.477	0.860
*III* vs. *II vs.0 + I*												

ER—estrogen receptor, PR—progesterone receptor, TNBC—triple negative BC, cT—clinic tumor, cN—clinic lymph node, pT—pathologic tumor, pN—pathologic lymph node, RCB—residual cancer burden, FR—fold regulation. The table shows the *p*-values for all the compared groups (as indicated in the first column) and where these differences were statistically significant, the expression level of that miRNAs (FR) in the interest group versus the reference group was calculated. According to data distributions, the differences in expression in case of two groups was evaluated with Mann–Whitney test and for three groups with Kruskal–Wallis test, followed by Dunn’s multiple comparison post hoc test. The *p*-values were highlighted in red when a significant value (<0.05) was obtained, and asterisks indicate the magnitude of *p*-value (* *p* < 0.05, ** *p* < 0.01, *** *p* < 0.001, **** *p* < 0.0001). In case of three or more groups, the asterisk next to the FR value is associated to the *p*-value obtained by the Dunn test.

**Table 4 ijms-23-12625-t004:** Associations of miRNAs expression in tumor tissue with clinical-pathological features and plasma sEVs.

	miR-21-5p	miR-221-3p	miR-146a-5p	miR-26a-5p
	*p-*Value	*p-*Value	*p-*Value	*p-*Value
**Age**	0.817	0.183	0.276	0.086
	R = −0.03	R = 0.19	R = −0.15	R = 0.24
**Grading Biopsy**	**0.045** *	0.435	0.052	0.679
*G3* vs. *G2 vs. G1*	FR G3 vs. G2 = 1.39 *			
	FR G2 vs. G1 = −1.58 *			
**Estrogen receptor**	0.344	0.304	**0.0003** ***	0.573
*ER+* vs. *ER−*			FR = −3.11	
**Progesteron receptor**	0.629	0.265	**0.042** *	0.454
*PR+* vs. *PR−*			FR = −2.02	
**HER2**	0.428	**0.034** *	0.061	**0.034** *
*HER2+* vs. *HER2−*		FR = 1.60		FR = 1.50
**KI67**	0.946	0.982	0.314	0.158
*>20* vs. *≤20*				
**Molecular Subtype**	0.395	0.641	**0.019** *	**0.042** *
*TNBC vs.*			FR TNBC vs. LumB = 3.81 **	FR TN vs. LumA = −1.87 *
*LuminalB* vs. *LuminalA*				
**cT**	0.191	0.907	0.123	0.875
*T4* vs. *T3 vs. T1 + T2*				
**cN**	0.065	0.377	0.382	0.767
*Positive (N1 + N2 + N3)* vs. *Negative (N0)*				
**Clinical Stage**	0.129	0.883	0.089	0.730
*High (S3 + S4)* vs. *Low (S1 + S2)*				
**pT**	0.648	0.151	0.074	0.561
*T2* vs. *T1 vs. T0*				
**pN**	0.655	0.444	0.318	0.100
*Positive (N1 + N2 + N3)* vs. *Negative (N0)*				
**Lymphatic Invasion**	0.586	0.864	0.735	0.139
*L1* vs. *L0*				
**MP** *G5* vs. *G4 vs. G3* vs. *G2 vs. G1*	0.149	**0.020** *	**0.008** **	0.230
		FR G5 vs. G3 = 2.80 ***	FR G5 vs. G3 = 5.00 ***	
		FR G5 vs. G1 = 1.58 *	FR G3 vs. G2 = −2.90 *	
			FR G3 vs. G1 = −2.73 *	
**RCB**	0.176	0.197	0.180	0.938
*III* vs. *II vs.0 + I*				
**Correlations between miRNAs expression in tumor tissue and sEVs**				
	0.781	0.421	0.543	0.953
	R = 0.04	R = 0.13	R = 0.09	R = −0.01

ER—estrogen receptor, PR—progesterone receptor, TNBC—triple negative BC, cT—clinic tumor, cN—clinic lymph node, pT—pathologic tumor, pN—pathologic lymph node, RCB—residual cancer burden, FR—fold regulation. The table shows the *p*-values for all the compared groups (as indicated in the first column) and where these differences were statistically significant, the expression level of that miRNAs (FR) in the interest group versus the reference group was calculated. According to data distributions, the differences in expression in case of two groups was evaluated with Mann–Whitney test and for three groups with Kruskal–Wallis test, followed by Dunn’s multiple comparison post hoc test. The *p*-values were highlighted in red when a significant value (<0.05) was obtained, and asterisks indicate the magnitude of *p*-value (* *p* < 0.05, ** *p* < 0.01, *** *p* < 0.001). In case of three or more groups, the asterisk next to the FR value is associated to the *p*-value obtained by the Dunn test.

**Table 5 ijms-23-12625-t005:** Candidate miRNAs target genes and processes.

Name	Genes	Adj *p*-Value *
**miR-21-5p**		
GO:0004674 Protein serine/threonine kinase activity	*GSK3B; MAPK6*	0
GO:0004842 Ubiquitin-protein transferase activity	*TOPORS; TRIM33; RMND5A*	0.0046
**miR-221-3p**		
R-HSA-9609507 Protein localization	*PEX1; CROT; PEX19*	0.013
R-HSA-74160 Gene expression (Transcription)	*ESR1; TFAP2A; SIRT1; MDM2; APAF1; DICER1; BCL2L11; ZKSCAN8; ZNF571; WDR61; ZFP30; ARID1A; CSTF2T*	0.0136
R-HSA-73857 RNA Polymerase II Transcription	*ESR1; TFAP2A; SIRT1; MDM2; APAF1; BCL2L11; ZKSCAN8; ZNF571; WDR61; ZFP30; ARID1A; CSTF2T*	0.0139
R-HSA-212436 Generic Transcription Pathway	*ESR1; TFAP2A; SIRT1; MDM2; APAF1; BCL2L11; ZKSCAN8; ZNF571; ZFP30; ARID1A*	0.0179
R-HSA-5663202 Diseases of signal transduction by growth factor receptors and second messengers	*ESR1; ERBB4; MDM2; BCL2L11; AP3B1; BRAP*	0.0335
GO:0006625 Protein targeting to peroxisome	*PEX1; CROT; PEX19*	0
GO:0031648 Protein destabilization	*SIRT1; MDM2; MYLIP*	0
GO:0006511 Ubiquitin-dependent protein catabolic process	*MDM2; MYLIP; RNF20; NDFIP1*	0.003
GO:0016567 Protein ubiquitination	*SIRT1; MDM2; WDR61; BRAP;* *MYLIP; RNF20*	0.0058
GO:0007399 Nervous system development	*ERBB4; APAF1; EVL; ARID1A; MYLIP*	0.0101
GO:0043065 Positive regulation of apoptotic process	*SIRT1; APAF1; BCL2L11*	0.0167
GO:0045893 Positive regulation of_transcription,_DNA-templated	*ESR1; TFAP2A; ERBB4; ARID1A; RNF20*	0.019
GO:0008285 Negative regulation of cell_population_proliferation	*TFAP2A; ERBB4; EIF2AK1; DDAH1; BECN1*	0.0196
GO:0045892 Negative regulation of transcription, DNA-templated	*TFAP2A; PHF12; SIRT1; MDM2*	0.0231
GO:0005777 Peroxisome	*PEX1; CROT; PEX19*	0
GO:0005778 Peroxisomal membrane	*PEX1; PEX19; ACSL3*	0
GO:0032991 Protein-containing complex	*ESR1; ACVR2B; MDM2; APAF1; PEX19; RBM39; SNX4*	0.0006
GO:0004842 Ubiquitin-protein transferase activity	*MDM2; HECTD2; BRAP; MYLIP; RNF20*	0.0002
GO:0061630 Ubiquitin protein ligase activity	*MDM2; BRAP; MYLIP; HECTD1*	0.0002
GO:0003677 DNA binding	*HMGXB4; TFAP2A; POGZ; DICER1; GLYR1; ARID1A; CERS2; PALB2*	0.0032
**miR-146a-5p**		
GO:0043235 Receptor complex	*ERBB4; LRP2*	0
**miR-26a**		
R-HSA-162582_Signal Transduction	*GSK3B; KPNA2; MAPK6*	0.05
GO:0098978 Glutamatergic synapse	*GSK3B; KPNA2; HSPA8*	0
GO:0004674 Protein serine/threonine kinase activity	*GSK3B; MAPK6*	0

* Benjamini-Hochberg (BH).

**Table 6 ijms-23-12625-t006:** Drug interaction predictions.

miRNA	Target Gene	Targeting Drug
hsa-miR-21-5p	*AP1*	topotecan
	*BCL2; PTEN; RECK*	gemcitabine
	*CDC25A*	3,3′-diindolylmethane
	*ICAM1; SERPINB5*	tretinoin
	*LRRFIP1*	teniposide
	*MSH2; PDCD2; PTEN; SPRY2*	5-fluorouracil
	*PDCD4*	arsenic trioxide, cytarabine, cisplatin, docetaxel, doxorubicin, paclitaxel,
	*PTEN*	curcumin, daunorubicin, doxorubicin, transtuzumab,
	*SPRY2*	metformin
hsa-miR-221-3p	*ABCB1*	trail
	*CDKN1A*	glucocorticoids
	*CDKN1B; ESR1*	tamoxifen
	*CDKN4*	gemcitabine
hsa-miR-146a-5p	*CFH*	pyrollidine dithiocarbamate, resveratrol
	*ERBB4*	doxorubicin

## Data Availability

Not applicable.

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
