# Peer review of "Circulating Small EVs miRNAs as Predictors of Pathological Response to Neo-Adjuvant Therapy in Breast Cancer Patients"

_ijms, 2022, doi:10.3390/ijms232012625_

Round 1

Reviewer 1 Report

This research article is focused on examining the potential predictive power of circulating microRNAs from plasma exosomes of breast cancer patients for the pathological response to neo-adjuvant therapy (NAT)  with bioinformatics tools and analysis. Four candidate miRNAs (miR-21-5p, 221-3p, 146a-5p and 26a-5p) associated with Miller-Payne (MP) pathological response to NAT were identified. Further analysis discloses that these miRNAs and their target genes can be modulated by different NAT regimens. 

This content of the paper is well organized and the study is presented in a logical order. The authors did a great job in categorizing and laying out patient and miRNA data. This paper contributes towards the potential application of exosomal miRNAs as predictive biomarkers in breast cancer.

A few minor changes need to be done to improve the manuscript,  as listed below:

1. No definition was provided for some abbreviations or terms when mentioned at first time in the text as well as in  the table such as “TNBC”, “TNM”, “TS-miRs”, ”G-CSF”, ”NSCLC”,  “RTE”, and “RCB”. Please go over the manuscript and add revise them.

2. Consider adding legend below Tables to make readers to easier  understand the Tables, say explanation of terms,  adding necessary descriptions for red color text, what the (c) and (p) stand for,etc.

3. There are some typos and grammatical errors in the text. Please carefully proofread the manuscript and correct them. For example, pathological complete response (pCR) instead of “complete pathological response” (line 60); KEGG instead of “KEEG” (line 183); some sentences are problematic ( line 90-91, line 109).

Author Response

Dear Reviewer, thank you very much for your observations. By addressing the answers to your observations we consider that this paper was clearly improved.

Best regards!

Dr. Ovidiu Balacescu

Reviewer 2 Report

This is a well written article describing an attempt to relate miRNA expression in extracellular vesicles (EVs) to predictors of response to treatment in breast cancer patients. The authors selected miRNA based on analysis of miRNA expression levels in cells (not EVs) and did a targeted analysis of the same in EVs precipitated from plasma with a commercial product. 

Unfortunately, the authors have not followed MISEV recommendations and did not demonstrate that their isolation method was isolating EVs in their hands by either TEM or western blot analysis of EV protein markers. In addition, EV number and size should also have been quantified. This is an essential component of any publication on EVs.

Secondly, the authors use the term "exosome" and "exosomal" but have not demonstrated that the EVs being studied are exosomes and not another type of EV. Along this line, the authors have not provided any introduction to EVs in the introduction section. 

For the qPCR, amplification efficiencies need to be provided for the miRNA that were able to be amplified. This is essential to know that the assay was indeed able to discriminate different amounts of miRNA.

The authors need to show the data from the amplification of the miRNA in a figure and not just the correlation analysis in a table. Both are necessary.

Author Response

Dear Reviewer, thank you very much for your observations. By addressing the answers to your observations we consider that this paper was highly improved.

Best regards!

Dr. Ovidiu Balacescu

Round 2

Reviewer 2 Report

The amplification efficiencies should be included in the article itself not just in the response to reviewers.

I suggest including the very informative NTA and TEM data in the paper. This is important information that could be included in the supplementary data (I was unable to open the .rar format).

Author Response

Dear Reviewer,

We answer your requests and consequently modify the manuscript as presented in the attached file.

Best regards!

Ovidiu Balacescu
